# A rubric for assessing conformance to the Ten Rules for credible practice of modeling and simulation in healthcare

Alexandra Manchel[1,2], Ahmet Erdemir[2,3], Lealem Mulugeta[2,4], Joy P. Ku[2,5], Bruno V. Rego[2,6], Marc Horner[2,7], William W. Lytton[2,8,9], Jerry G. Myers[2,10], Rajanikanth Vadigepalli[1,2]*

**1** Department of Pathology and Genomic Medicine, Thomas Jefferson University, Philadelphia, Pennsylvania, United States of America, **2** Committee on Credible Practice of Modeling, & Simulation in Healthcare, Interagency Modeling and Analysis Group and Multiscale Modeling Consortium, Bethesda, Maryland United States of America, **3** Department of Biomedical Engineering, Lerner Research Institute, Cleveland Clinic, Cleveland, Ohio, United States of America, **4** InSilico Labs LLC, Houston, Texas United States of America, **5** Department of Bioengineering, Stanford University, Stanford, California, United States of America, **6** Department of Biological & Agricultural Engineering, Louisiana State University, Baton Rouge, Louisiana, United States of America, **7** Ansys Inc. Evanston, Evanston, Illinois, United States of America, **8** Downstate Health Sciences University, Brooklyn, New York, United States of America, **9** Kings County Hospital, Brooklyn, New York, United States of America, **10** NASA - John H Glenn Research Center, Cleveland, Ohio, United States of America

\* rajanikanth.vadigepalli@jefferson.edu

## Abstract

The power of computational modeling and simulation (M&S) is realized when the results are credible, and the workflow generates evidence that supports credibility for the context of use. The Committee on Credible Practice of Modeling & Simulation in Healthcare was established to help address the need for processes and procedures to support the credible use of M&S in healthcare and biomedical research. Our community efforts have led to the Ten Rules (TR) for Credible Practice of M&S in life sciences and healthcare. This framework is an outcome of a multidisciplinary investigation from a wide range of stakeholders beginning in 2012. Here, we present a pragmatic rubric for assessing the conformance of an M&S activity to the TR. This rubric considers the ability of an M&S study to communicate how well the study conforms to the Ten Rules for credible practice and facilitate outreach to a wide range of stakeholders from context-specific M&S practitioners to policymakers. It uses an ordinal scale ranging from Insufficient (zero) to Comprehensive (four) that is applicable to each rule, providing a uniform approach for comparing assessments across different reviewers and different modeling studies. We used the rubric to evaluate the conformance of two computational modeling activities: 1. six viral disease (COVID-19) propagation models, and 2. a model of hepatic glycogenolysis with neural innervation and calcium signaling. These examples were used to evaluate the applicability of the rubric and illustrate rubric usage in real-world M&S scenarios including those that

**Data availability statement:** All relevant data are within the paper and its Supporting information files.

**Funding:** Contributions by Rajanikanth Vadigepalli were supported by the National Institutes of Health under Grants R01AA018873, R01HL161696, and OT2OD030534. Contributions by Alexandra Manchel were supported by the National Institutes of Health under Grant F31AA030214. Contributions by Ahmet Erdemir were supported in part by the National Institutes of Health under Grant R01EB024573. Joy Ku was supported by the National Institutes of Health under Grants P2C HD101913, R01GM124443, and P41 EB027060. Contributions by William W Lytton were supported by National Institutes of Health under Grants OT20D03054 and R01MH086638. The funders had no role in the study design and implementation, interpretation of the results and preparation of the manuscript.

**Competing interests:** We have read the journal's policy and the authors of this manuscript have the following competing interests: Ahmet Erdemir owns and operates innodof, LLC, a consulting company for modeling and simulation. Lealem Mulugeta owns and operates InSilico Labs LLC and Medalist Performance. InSilico Labs provides computational modeling and simulation products and services, and Medalist Performance applies computational and biomedical approaches to provide peak performance coaching services to tactical professionals, athletes, astronauts, and executives. Marc Horner is employed by ANSYS, Inc., a company that develops commercial off-the-shelf computational modeling software. The remaining authors have declared that no competing interests exist. This does not alter our adherence to PLOS ONE policies on sharing data and materials.

bridge scientific M&S with policymaking. The COVID-19 M&S studies were of particular interest because they needed to be quickly operationalized by government and private decision-makers early in the COVID-19 pandemic and were accessible as open-source tools. Our findings demonstrate that the TR rubric represents a systematic tool for assessing the conformance of an M&S activity to codified good practices and enhances the value of the TR for supporting real-world decision-making.

## Introduction

The role of computational modeling and simulation (M&S) in healthcare research and clinical practice is expanding at a rapid pace. M&S approaches have been integral to the progress in biomedical sciences and are starting to enable *in silico* and systems medicine efforts [1,2]. Computational modeling is relatively new in clinical and biomedical settings, necessitating the standardization of M&S efforts. The addition of standardized practices increases the credibility of the practice of M&S in this area as it has done in other disciplines, such as engineering. This also increases the M&S usefulness and widespread adaptation. Multiple standards have been proposed, both in industry and government, for establishing and ensuring credibility of M&S practices in various engineering fields, including medical devices [3–7]. Similarly, multiple standards exist for systems biology applications, which have been reviewed in Tatka et al. 2023 [8], that aim to address conceptual information, nomenclature, data formats, and representations of biochemical systems, and intend to improve the communication and sharing of M&S components.

In order to promote this standardization process in the biomedical community beyond those working in systems biology, the IMAG/MSM Committee on Credible Practice of Modeling and Simulation in Healthcare developed the "Ten Rules for Credible Practice of Modeling and Simulation in Healthcare" [9]. This framework is an outcome of multidisciplinary input from a wide range of stakeholders [10,11]. These rules aim to establish a unified conceptual framework to design, implement, evaluate, and communicate the activities, products, and outcomes of M&S in the biomedical sciences and clinical care domain. In application, the unified framework enables outreach to the entire M&S user community, ranging from model developers to policy makers to clinicians and other non-M&S practitioners.

Recently, Tatka et al. [8] reviewed the existing standards for representing and documenting systems biology models. Current standards are limited to agreed-upon modeling formats as a means to share information; however, there is no widely utilized standard for assessing credibility of the practice in this area. As Tatka et al. [8] noted in their review, standards for model annotation must become more widely accepted such that interoperability, reusability, comparability, and comprehension can be improved. Credible practice will also be enhanced when the information needed for simulation and parameter estimation is explicitly defined and stated. Lastly, reproducibility would not be possible without efficient dissemination of all artifacts and proper documentation on an open-source repository platform. When an M&S study

conforms to credible modeling practice guidelines at a high level, there is outreach to a wide range of stakeholders, thus with proper dissemination of documentation, one will largely be able to independently reproduce the M&S results. The authors of Tatka et al. [8] note that there is a lack of consensus on quantitative credibility scoring and that a system that addresses this area would provide the community of practice with a metric for comparing the credibility of models and a guide for the development of more credible models.

The Committee's Ten Rules for Credible Practice of M&S in Healthcare (Table 1) establishes initial standards for systems modeling and beyond [9]. While every effort was made to thoroughly describe and define the rules, it lacks a quantitative, rigorous, and repeatable metric. A consistent application of the rules likely requires a complementary rubric for assessing conformance to the rules and evaluating the credibility of the M&S practice. Such a rubric would be used to assess and communicate various aspects of the Ten Rules (TR), including the validity, level of detail, and overall "correctness" of the M&S practice.

In principle, a case can be made that each of the Ten Rules in Table 1 needs its own assessment approach. For example, Rule 1 on defining the context of use can be assessed according to factors that quantify the level of detail in the documentation of the M&S subject, scope of the M&S purpose or results and intended use of the M&S results such as to support clinical decision making, inform regulatory evidence, or to inform next research steps. Additionally, Rule 3 on model evaluation requires extensive consideration of how the M&S activity and results are verified and validated, as well as how the assessment is presented to support the intended use. This rule is consistent with U.S. Food and Drug Administration (FDA) guidance and American Society of Mechanical Engineers (ASME) standards for best practices in verification and validation of medical devices [4,12]. However, such a customized, rule-specific assessment approach might become overly complex and unwieldy for consistently evaluating conformance to the Ten Rules.

To address this issue, the Committee formulated a rubric based on the ability of the M&S to facilitate outreach to a wide range of stakeholders from context-specific M&S practitioners to policymakers. Since there exist various levels of M&S expertise in the healthcare domain, the need for direct and clear communication of M&S results is essential. The development of the Ten Rules rubric facilitates such communication and understanding of computational modeling implementation

**Table 1. The Committee's Ten Rules of credible practice of M&S in healthcare [9].**

| Rule | | Description |
|---|---|---|
| 1. | Define context clearly | Develop and document the subject, purpose, and intended use(s) of the model or simulation |
| 2. | Use contextually appropriate data | Employ relevant and traceable information in the development or operation of a model or simulation |
| 3. | Evaluate within context | Perform verification, validation, uncertainty quantification, and sensitivity analysis of the model or simulation with respect to the reality of interest and intended use(s) of the model or simulation |
| 4. | List limitations explicitly | Provide restrictions, constraints, or qualifications for or on the use of the model or simulation for consideration by the users or customers of a model or simulation |
| 5. | Use version control | Implement a system to trace the time history of modeling and simulation activities including delineation of each contributors' efforts |
| 6. | Document appropriately | Maintain up-to-date informative records of all modeling and simulation activities, including simulation code, model mark-up, scope and intended use of modeling and simulation activities, as well as users' and developers' guides |
| 7. | Disseminate broadly | Share all components of modeling and simulation activities, including simulation software, models, simulation scenarios and results |
| 8. | Get independent reviews | Have the modeling and simulation activity reviewed by nonpartisan third-party users and developers |
| 9. | Test competing implementations | Use contrasting modeling and simulation implementation strategies to check the conclusions of different strategies against each other |
| 10. | Conform to standards | Adopt and promote generally applicable and discipline specific operating procedures, guidelines, and regulations accepted as best practices |

and simulation results between stakeholders. This includes, but is not limited to, communication between the model developers, M&S practitioners, model end-users, as well as clinicians, policy makers, and other decision makers who depend on the knowledge generated by the M&S. Therefore, the purpose of the rubric is to evaluate and assess an M&S study's conformance to credible practices outlined in the Ten Rules as it pertains to outreach capability.

Most recently, the global response to the COVID-19 pandemic highlights the need for a systematic assessment of credible practice of M&S across this entire spectrum of stakeholders [13–17]. The role of M&S in providing quantitative insight for COVID-19 spread in the general population was called into question due to a failure to predict early (circa 2020) outbreak dynamics [13]. Nonetheless, the model predictions strongly influenced decision makers due to the ability of M&S practitioners to quickly generate results with a perceived to be high degree of precision superior to available observational statistical analyses. In retrospect, although results were computationally precise, they exhibited lower accuracy than initially anticipated. Model transparency, which includes explicit documentation of model choices, assumptions, the steps in the modeling process, and the expectations for the outputs, provides a reasonable defense against the propagation of misinformation and misunderstanding, such as what occurred during the pandemic [15–16]. In several instances during the pandemic, a model developed for population level COVID-19 spread in a large geographical region was applied to a less appropriately applicable region without significant tuning and modification to account for population-specific demographic, clinical, epidemiological, and other influencing factors [18]. Such activities during the early part of the COVID-19 pandemic illustrate how the lack of transparency and independent evaluation reduces the utility of models to inform critical decisions [17]. Informing the user of the M&S results of model context and its intended use, such as through the assessment of its conformance to the "Ten Rules in Healthcare" will greatly minimize the negative impacts on model utility at all levels of application [9]. Such an assessment would not just improve transparency but would enable communication of credible practice of M&S in a comprehensive manner (Table 1; [9]).

The remainder of the manuscript is organized as follows: First, we describe our process for developing and utilizing the rubric. We then present the rubric and an explanation of its components. Next, we illustrate the application of the rubric in multiple use cases to evaluate its utility in assessing the conformance to the Ten Rules in a consistent manner across multiple reviewers and M&S studies. Finally, we discuss best practices for applying the rubric and possible future extensions.

## Materials and methods

### Development of the rubric framework

Our proposed rubric can be used to assess and communicate the extent of conformance to the Ten Rules for Credible Practice of M&S based on the capability of outreach to the biomedical and healthcare community (Table 2). The concept of using a rubric for communicating the credibility state of an M&S evolved from challenges in communicating the ten

**Table 2. Rubric for assessing conformance to the Ten Rules.**

| Outreach Capability | Outreach to application-domain experts who may not be M&S practitioners | Outreach to M&S practitioners who may not be application-domain experts | Outreach to application-domain specific M&S practitioners | Outreach to application-domain specific M&S practitioners | None or very limited |
|---|---|---|---|---|---|
| **Conformance Level** | Comprehensive | Extensive | Adequate | Partial | Insufficient |
| | 4 | 3 | 2 | 1 | 0 |
| **Description Level** | Can be understood by non-M&S practitioners familiar with the application domain and the intended context of use | Can be understood by M&S practitioners not familiar with the application domain and the intended context of use | Can be understood by M&S practitioners familiar with application domain and the intended context of use | Unclear to the M&S practitioners familiar with the application domain and the intended context of use | Missing or grossly incomplete information to properly evaluate the conformance with the rule |

simple assessments at different decision-making levels and in different contextual applications. Thus, the intent of the rubric development, is to bring a concise communication tool to the M&S healthcare community.

To develop the rubric, the Committee considered requirements of outreach to a wide range of stakeholders (Fig 1A), each of whom has their own distinct use cases and priorities in evaluating an M&S model. For instance, M&S practitioners may want to conduct granular analysis of their own M&S practices, while clinicians are primarily concerned with whether they can trust M&S to inform a clinical practice decision.

The Committee developed the rubric framework through an iterative approach. The initial framework had reviewers assess models qualitatively, ranking the conformance of a model to each of the Ten Rules as insufficient, partial, adequate, extensive, or comprehensive (Table 2). The qualitative assessment made it challenging to compare reviewer assessments and derive an overall rating for the model when there existed variability between the individual reviewers' assessments, as in the case of the first COVID-19 model (UPenn's COVID-19 model) to which the Committee applied the rubric. Therefore, a second development of the Ten Rules rubric was implemented. In this development, a scoring system was included such that for each rule, the level of conformance is given a numerical score.

## Application of the rubric for different use cases

We applied the Ten Rules and the rubric to evaluate the M&S practices of several COVID-19 modeling studies with versions released early in the pandemic and available at the time of this study: MIT model [19], IHME model [20,21], CU model [22], NE model [23], ICL model [24], UPenn model [25]. Model details can be found in Table 3. Independent reviews by persons with significant experience in M&S credibility assessment and with some familiarity of the application of M&S in supporting government and medical industry decision making are used to assess each model.

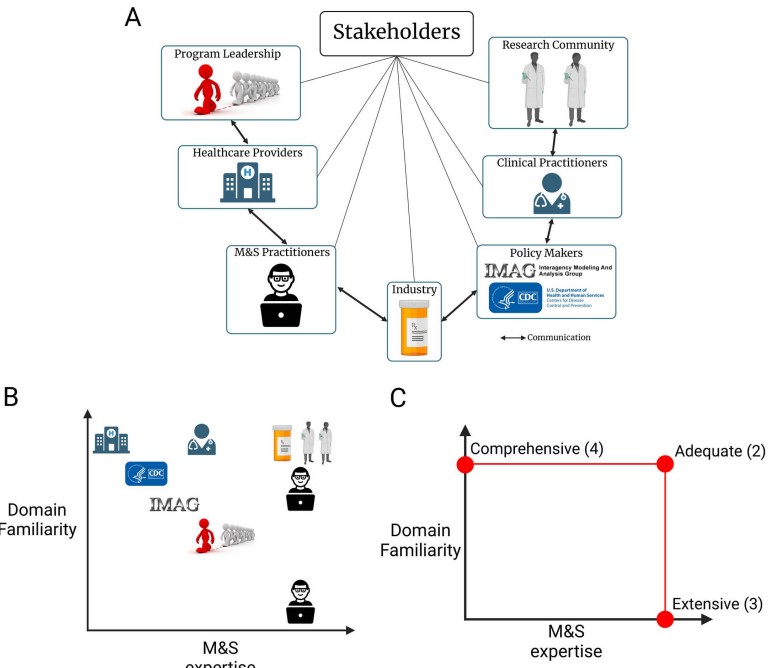

**Fig 1. Assessing TR rubric conformance to the Ten Rules based on the extent of outreach to the stakeholder's domain expertise.** (A) The range of stakeholders that may utilize the rubric. (B) Example distribution of the stakeholders in the stakeholder assessment chart. (C) The conformance levels to each of the Ten Rules based on the extent of outreach to stakeholders along the M&S expertise and domain familiarity axes.

**Table 3. COVID-19 models tested for their conformance to the Ten Rules.**

| Location of Model Development | Model Description | Website | Dates Accessed | References |
|---|---|---|---|---|
| Massachusetts Institute of Technology (MIT) | A novel epidemiological model for predicting detected cases and deaths in the pre-vaccination era of the COVID-19 pandemic | https://www.covidanalytics.io/ | June 2020 September 2020 February 2021 | [19] |
| Institute for Health Metrics and Evaluation (IHME) | A model for predicting possible trajectories of COVID-19 infections and the effects of non-pharmaceutical interventions in the United States | https://www.healthdata.org/covid | June 2020 September 2020 February 2021 | [20,21] |
| Columbia University (CU) | A model to infer critical epidemiological characteristics associated with COVID-19 | https://columbia.maps.arcgis.com/apps/webappviewer/index.html?id=ade6ba-85450c4325a12a5b9c09ba796c | June 2020 September 2020 February 2021 | [22] |
| Northeastern University (NE) | A model used to study spatiotemporal COVID-19 spread | https://covid19.gleamproject.org/#model | June 2020 September 2020 February 2021 | [23] |
| Imperial College London (ICL) | A model used to study the effect of non-pharmaceutical interventions in controlling the COVID-19 epidemic | https://www.imperial.ac.uk/mrc-global-infectious-disease-analysis/disease-areas/covid-19/covid-19-planning-tools/ | June 2020 September 2020 February 2021 | [24] |
| University of Pennsylvania (UPenn) | COVID-19 Hospital Impact Model for Epidemics (CHIME), which was designed to assist hospitals and public health officials with understanding hospital capacity needs during the pandemic | https://penn-chime.phl.io/ | April 2020 June 2020 September 2020 February 2021 | [25] |

The selection process for the COVID-19 models we evaluated did not take into account potential conformance to the Ten Rules, but rather addresses the availability of the model and related information present at the time of the study. As none of the models explicitly followed our recommended credibility practice, we did not seek to compare the COVID-19 models to determine which one is the most conformant to the Ten Rules but rather to express our assessment based on the information provided. Finally, this is not an endorsement or criticism of the M&S practices utilized for the models tested for conformance to the Ten Rules. Instead, we seek to exemplify how the rubric is to be employed when testing for M&S conformance and outreach. Additionally, we note that no attempts were made to reproduce any of the results reported by these models, nor was any attempt made to assess the scientific validity of the models, assumptions, or limitations. Instead, we assessed the outreach capability of the models and were interested in their representation and dissemination.

Two independent reviewers evaluated the conformance of the UPenn COVID-19 model (details can be found in Table 3) to the Ten Rules in April 2020. They used the initial, qualitative rubric. A separate independent reviewer (Reviewer 3) assessed the conformance of the remaining COVID-19 models to the Ten Rules using the numerical scoring version of the rubric. Reviewer 3's original assessment was performed on June 24, 2020 and repeated twice: once on September 7, 2020 and again on February 5, 2021.

Finally, the rubric was applied in a clinically relevant M&S study to evaluate the utility of our approach in assessing the extent of credibility of M&S practices in these contexts of use. In the study, the authors developed a multi-scale, multi-organ model of hepatic metabolism. The authors performed a self-assessment of their model's conformance to the Ten Rules prior to their initial manuscript submission on this model and then reassessed their model during the manuscript revision process.

## Results

### A generalized rubric based on outreach capability

The Committee recognized that the rubric needed to account for the different stakeholders who may be interested in utilizing a model. Assessment of a model's ability to communicate how and if it satisfied each of the Ten Rules would differ greatly depending on the stakeholder. Therefore, in the proposed rubric, the key stakeholder traits are distributed along two axes: their level of M&S expertise and their familiarity with the biological domain (Fig 1B). The stakeholder communities can have different mixtures of M&S expertise and domain familiarity. Individuals with expertise in M&S and the biological domain relevant to the context of use are positioned towards the upper right, while individuals with very little M&S expertise and domain knowledge are positioned towards the lower left. The rubric assesses the conformance to each of the Ten Rules based on the extent of outreach to each group (Fig 1C). For a given rule, if the M&S practice was conducted at a level that is primarily accessible to only those with M&S expertise and domain familiarity, we deem this practice to be conformant to the rule at the Adequate level. If the M&S practice of a given rule is more broadly understood by individuals with M&S expertise without familiarity of the specific biological domain, we deem this practice to be conformant to the rule at the Extensive level. If the M&S practice of a given rule is understood by those familiar with the biological domain but do not have M&S expertise, we deem this practice to be conformant to the rule at the Comprehensive level. Lastly, the M&S practice that is unclear to the M&S practitioners with familiarity of the biological domain is considered as a Partial level of conformance, with missing information assessed as an Insufficient level.

In this rubric, the model with the highest conformance level (Comprehensive) provides outreach to domain experts who may not be M&S practitioners while the lowest conformance level (Insufficient) does not provide sufficient outreach to any community level. Taken together, this rubric provides a generalized and graded approach to assess the conformance to the Ten Rules (Fig 1C). Table 2 shows a concise representation of the proposed conformance rubric to the Ten Rules of credible practice of M&S in healthcare. The extended rubric can be found in S1 File.

The rubric does not assess the "correctness" (i.e., the validity or accuracy) of the computational models, but rather analyzes M&S credible practice conformance based largely on two dimensions: M&S experience and scientific domain expertise (Fig 1). We note that the rubric for conformance to the Ten Simples Rules is not an M&S practice accreditation process, but rather a communication tool for analyzing the robustness of the M&S practice employed for a computational model within specifically stated context of use.

### Assessment criteria for each rule

Within each of the Ten Rules (i.e., guiding principles of M&S practice), we specified the detailed criteria to assess the level of conformance and outreach capability to all stakeholders across different application contexts (S1 File). For instance, Rule #1 is to define the context of use clearly by developing and documenting the application, purpose, and intended uses of the model and simulation (Table 1). In our proposed rubric, an M&S practice conforms to this rule at the highest level (Comprehensive) if:

1. a summary of the context definition can be understood by non-M&S practitioners,

2. detailed explanation is understandable by experts from the application domain that may not be M&S practitioners, and

3. many relevant details are included in the documentation that enable adequate understanding by both application domain-specific and non-domain M&S experts.

The next levels of conformance are based on whether the context definition was communicated at a level that is a) understandable only by M&S experts, even if they are from outside of the application domain (Extensive); b) restricted to M&S experts with experience in the specific application domain (Adequate); or c) achieved only partially (Partial).

As another example, Rule #2 is to use contextually appropriate data by employing relevant and traceable information in the development or operation of a model or simulation (Table 1). In our proposed rubric, M&S practice conforms to this rule at the highest level (Comprehensive) if:

1. the data used in the M&S development is contextually appropriate,

2. all the data used in M&S development and/or operation is traceable to its original source, and

3. application-domain experts that are not M&S practitioners can understand which and how the data was used.

As a general guideline, the level of conformance of an M&S practice to each rule should be assessed systematically by answering the following three questions:

1. Does the M&S practice employ the rule as defined by Erdemir et al. (2020)?

2. What is its outreach capability, or which types of stakeholders can the M&S practice effectively support?

3. How easily can different stakeholders understand the extent to which the rule was applied to the M&S practice?

It is also important to note that if the answer to the first question is "no" or "uncertain," the conformance level of the M&S practice for that rule must be scored as 0 (Insufficient). For example, in the case of Rule #2, if the data used in the M&S practice is entirely inappropriate, the answer to the first question would be "no." As a result, the conformance score for Rule #2 would be 0, regardless of how traceable or well-documented the data is in that M&S practice.

The detailed criteria corresponding to all of the Ten Rules can be found in S1 File.

**Numerical scoring used in the rubric**

A numerical scoring system was included in the rubric to quantify the assessments with each level of conformance. A conformance level of Insufficient is given a score of 0, while a conformance level of Comprehensive is given a score of 4.

After the reviewer has completed their assessment of the model's conformance to the rules, a total numeric score can be calculated, thereby allowing for a higher-level understanding of the model's conformance and providing a means of easily comparing assessments between reviewers. Assessment of a model which reaches an overall conformance level of Comprehensive will have a total score in the range [35, 40], while a model with Insufficient conformance will have a total score in the range [0, 5)) (Table 4). The total score is most informative at the high and low extremities, as the M&S study's overall performance can be easily evaluated

If a subset of the TR is not included in a model assessment, the overall scores associated with a given conformance level will need to be adjusted. For example, if two Rules are omitted in the assessment, a Comprehensive conforming model will then have a total score in the range [28, 32], rather than [35, 40]. Similarly, Extensive will have a score in the range [20, 28), Adequate will have a score in the range [12, 20), Partial will have a score in the range [4, 12), and Insufficient will have a score in the range [0, 4).

**Table 4. Numerical scoring system for assessing conformance to the Ten Rules.**

| Conformance Level | Score for each Rule | Score Range for Averaging across Rule | Score Range for Summing all Ten Rules |
|---|---|---|---|
| Comprehensive | 4 | [3.5, 4] | [35, 40] |
| Extensive | 3 | [2.5, 3.5) | [25, 35) |
| Adequate | 2 | [1.5, 2.5) | [15, 25) |
| Partial | 1 | [0.5, 1.5) | [5, 15) |
| Insufficient | 0 | [0, 0.5) | [0, 5) |

Numerical scoring also enables the calculation of statistics, such as averages and standard deviations, across multiple assessments for a single rule. When averaging scores across reviewers for a single rule, the score may not be a whole number, which we have accounted for in Table 4, which details the range of conformance scores for each rule.

### Recommended process for implementing rubric

The recommended process for implementing the rubric throughout the M&S life cycle begins with clearly identifying the M&S intended context of use, including M&S domain of use, use capacity, and strength of influence (Fig 2) [9]. Next, the conformance threshold must be established according to the rubric and TR. It is expected that throughout the M&S lifecycle there is to be further development and refinement of the model, thereby necessitating evaluation of the updated M&S per the Ten Rules and rubric thresholds. Following this assessment, there should be clear documentation and then implementation of the M&S. Additionally, when implementing the M&S activities, further reporting and documentation may be needed.

### Illustrative application of the rubric to assess COVID-19 M&S practice

We applied the Ten Rules and the rubric to evaluate the M&S practices of several COVID-19 modeling studies, as described in the Methods. Table 5 illustrates a summary of our two independent reviewer processes to evaluate the conformance to the Ten Rules of the UPenn COVID-19 model in April 2020. The complete assessment and conformance testing made by Reviewer 1 can be found in S2 File, and by Reviewer 2 in S3 File. Briefly, Reviewer 1 and 2 disagreed on the conformance level of the UPenn COVID-19 model for five of the ten rules (Rule #1, #2, #4, #8, and #10). Despite this variability, the overall conformance of the model as tested using the numeric scoring system resulted in Reviewer 1's overall score of 21 and Reviewer 2's overall score of 20. Both reviewers agreed that the overall conformance of the model was Adequate in that the model can be understood by those with expertise in M&S and the biological domain. Thus, the scoring system facilitates comparisons between reviewers, assessments for each individual rule, and also a model's

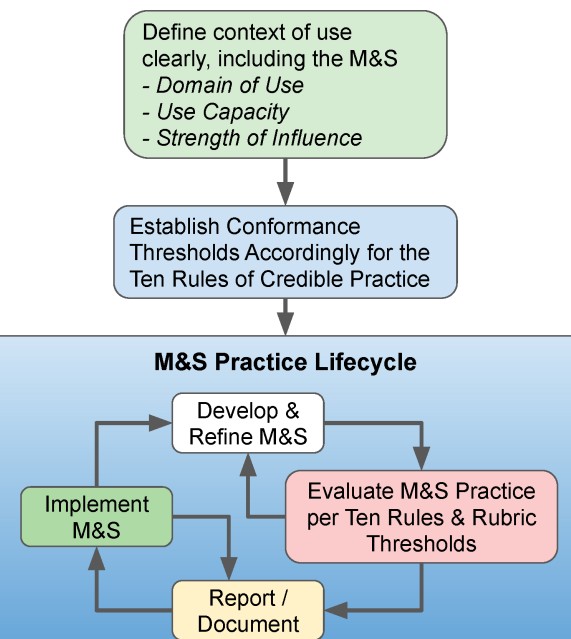

**Fig 2. Recommended process for implementing the TR rubric throughout the M&S lifecycle.**

**Table 5. Independent reviewer assessment of the UPenn COVID-19 model's conformance to the Ten Rules.**

| Rule | | Conformance Level | | Same Conformance across Reviewers? |
|---|---|---|---|---|
| | | Reviewer 1 | Reviewer 2 | |
| 1. | Define context clearly | Comprehensive | Adequate | No |
| 2. | Use contextually appropriate data | Adequate | Partial | No |
| 3. | Evaluate within context | Insufficient | Insufficient | Yes |
| 4. | List limitations explicitly | Adequate | Partial | No |
| 5. | Use version control | Extensive | Extensive | Yes |
| 6. | Document appropriately | Extensive | Extensive | Yes |
| 7. | Disseminate broadly | Comprehensive | Comprehensive | Yes |
| 8. | Get independent reviews | Insufficient | Partial | No |
| 9. | Test competing implementations | Partial | Partial | Yes |
| 10. | Conform to standards | Adequate | Comprehensive | No |

overall conformance. In the present rubric, the total assessment acts like an average, rather than a sum, of the individual rule assessments and is presented as representing an overall conformance using the same category scale as used for each individual rule.

A separate independent reviewer (Reviewer 3) assessed the conformance of the remaining COVID-19 models to the Ten Rules. For each of these model assessments, Rule #3 (Evaluate within context) and Rule #9 (Testing competing implementations) were not examined as these rules posed the greatest uncertainty and required in-depth knowledge of the model structure and development. Here, for simplicity, we discuss the results from the most recent assessment on February 5, 2021; however, extensive commentary from each assessment is documented in S4 File. The most recent assessment of the MIT COVID-19 model's conformance to the Ten Rules is exemplified in Table 6, and the reviewer's reasoning for each conformance score is highlighted in the Comments column of the table. Despite the reviewer-assessed overall model conformance level being Extensive, the total score was 18–19, which would point towards an overall conformance level of Adequate. The scoring system introduced in the Methods section provides reviewers with the ability to properly evaluate their complete assessment of model conformance to the Ten Rules in a more quantitative manner. Reviewer 3 followed the same protocol for assessing conformance of the remaining COVID-19 models to the Ten Rules. The models' conformances to the Ten Rules can be found in Table 7 and the detailed assessments can be found in S4 File. The overall conformance across the COVID-19 models assessed by Reviewer 3 were between Adequate and Extensive. All of the COVID-19 models have shown improvement according to the Ten Rules rubric following reassessment.

Following Reviewer 3's assessment of the five COVID-19 model conformances to the Ten Rules, we analyzed the results to identify the rules to which the models were least conformant. If a model was between two conformance levels for a given rule, the lower conformance level was used in the comparative analysis. The average numerical conformance score across all models for each rule was assessed. Those with an average score lower than 1.5 meant that the conformance level was at most Partial. The only rule that met this requirement was Rule #10: Conform to standards. In all COVID-19 models assessed by Reviewer 3, the conformance levels of the models to Rule #10 was either Partial or Insufficient, meaning the M&S practice of conforming to standards was incompletely stated (Partial conformance score) or insufficient evidence was presented to support conformance to standards (Insufficient conformance score) as assessed by M&S practitioners familiar with the application domain and the intended context of use. In order for the COVID-19 models to increase their conformance levels to Rule #10 of the Ten Rules, the models should have been implemented using community standards and formats. The associated documentation should lay out the details on the standards including version numbers and any exceptions or deviations that influence the use of the model. For instance, the IHME COVID-19 model is written in the widely used Python programming language; however, the model has not yet been configured for

**Table 6. Reviewer 3's assessment of the MIT COVID-19 model's conformance to the Ten Rules.**

| Rule | | Conformance Level | Conformance Level Score | Comments |
|------|---|---|---|---|
| 1. | Define context clearly | Extensive | 3 | • A critical tool for COVID-19 planning that charts the progression of the pandemic across the United States and the world. |
| 2. | Use contextually appropriate data | Extensive | 3 | • Country-level projections are modeled based on historical data to increase the accuracy of future predictions.<br>• Data is aggregated over 100 published clinical studies and preprints released between December 2019 and March 2020.<br>• Declaration of collaborators and partner institutions that provide data and insights to model development team. |
| 4. | List limitations explicitly | Adequate | 2 | • Differences between Johns Hopkins University map and MIT cases reported<br>• Total counts only account for countries in which they have sufficient data and where the pandemic is currently active.<br>• Limitations explicitly stated in the paper |
| 5. | Use version control | Extensive | 3 | • Extensive documentation on differences between versions and updates.<br>• Model codes are available on Github: https://github.com/COVIDAnalytics/DELPHI |
| 6. | Document appropriately | Extensive | 3 | • Model documentation contains the system of equations and rates.<br>• Code is well documented and there are detailed instructions on how to reproduce the results. |
| 7. | Disseminate broadly | Extensive | 3 | • Model results regularly published with interactive graphics.<br>• Results, data, models and simulations are openly available to the public and scientific community. |
| 8. | Get independent reviews | Insufficient/Partial | 0-1 | • Paper with scientific findings published in [19] |
| 10. | Conform to standards | Partial | 1 | • Codes are written in Python and Mathematica and data is provided in comma-separated variables (CSV) format. |
| **Overall Conformance** | | **Extensive** | **18-19** | • The epidemiologically based model is disseminated broadly and provides links to model descriptions and data sources from their project website.<br>• The code utilized in the research is accessible to the public via a GitHub repository and the model limitations are adequately described. |

use outside of the internal IHME infrastructure. The current Partial conformance to Rule 10 can be increased to Adequate and possibly Extensive if IHME provides sufficient evidence for following appropriate standards such as Python style guides and statistical modeling standards (e.g., The American Statistical Association's Ethical Guidelines for Statistical Practice).

### Illustrative application of the rubric to M&S of liver metabolism

The rubric was applied to a clinically relevant M&S study by Verma et al. [26] in which the authors developed a multi-scale, multi-organ model of hepatic metabolism. The authors performed a self-assessment of their model's conformance to the Ten Rules prior to their initial manuscript submission and then re-assessed their model during the manuscript revision process. Below is a summary of the author's self-assessment included with the manuscript as a way to illustrate the application of the rubric.

**Rule #1 (Define context clearly):** The authors provided a detailed description of the model's context written using terminology familiar to non-M&S practitioners who are knowledgeable about the application domain, so the authors described the model's conformance to Rule #1 (Define context clearly) as Comprehensive (score = 4). Briefly, the authors described that the primary goal of the model was to evaluate the role of neural signals in controlling the metabolic functionality of the liver, particularly in regulating the glycogenolysis to maintain appropriate responses to hormonal signals

**Table 7. Reviewer 3 assessment of COVID-19 model conformances to the Ten Rules.**

| Rule | | Conformance Level (Score) | | | | |
|------|------|------|------|------|------|------|
| | | **IHME Model** | **CU Model** | **NU Model** | **ICL Model** | **Average Conformance across models** |
| 1. | Define context clearly | Extensive (3) | Extensive (3) | Extensive (3) | Adequate/ Extensive (2–3) | Extensive (2.75) |
| 2. | Use contextually appropriate data | Extensive (3) | Extensive (3) | Adequate/ Extensive (2–3) | Adequate/ Extensive (2–3) | Extensive (2.5) |
| 4. | List limitations explicitly | Partial (1) | Adequate (2) | Adequate (2) | Adequate (2) | Adequate (1.75) |
| 5. | Use version control | Adequate/ Extensive (2–3) | Adequate (2) | Insufficient (0) | Extensive (3) | Adequate (1.75) |
| 6. | Document appropriately | Adequate (2) | Adequate (2) | Extensive (3) | Extensive (3) | Extensive (2.5) |
| 7. | Disseminate broadly | Adequate/ Extensive (2–3) | Adequate/ Extensive (2–3) | Adequate/ Extensive (2–3) | Adequate/ Extensive (2–3) | Adequate (2) |
| 8. | Get independent reviews | Adequate/ Extensive (2–3) | Extensive (3) | Adequate (2) | Extensive (3) | Extensive (2.5) |
| 10. | Conform to standards | Partial (1) | Partial (1) | Insufficient (0) | Partial (1) | Partial (0.75) |
| **Overall Conformance** | | **Adequate (16)** | **Adequate (18)** | **Adequate (14)** | **Adequate (18)** | **Adequate (16.5)** |

to meet the systemic glucose demands. The biological domain, structures, spatial scales, and time scales are explicitly stated. Additionally, the authors included an explanation of other uses for the model, which include exploration of the effect of dietary intake and insulin resistance in promoting a hepatic steatosis-like phenotype in the context of innervation, calcium signaling and central nervous system (CNS) activation.

**Rule #2 (Use contextually appropriate data):** The authors believed their model conformed to an Extensive (score = 3) level for Rule #2 since the in vitro, ex vivo, in vivo preclinical and human subject data used for model building and validation was confirmed to meet the detailed data requirements for consistency and explicit description of data heritage.

**Rule #3 (Evaluate within context):** The authors' self-assessed conformance level was Extensive (score = 3) since verification and validation of the model output was explicitly described and performed by both the developer and a third-party lab member not involved in the study. Additionally, the authors state that during the revision process, there was extensive validation performed as the model was recalibrated based on experimental hepatic calcium dynamics and catecholamine secretion in humans during periods of increased exercise.

**Rule #4 (List limitations explicitly**): The model's conformance was considered to be Comprehensive (score = 4) as all limitations were explicitly provided. In addition, the limitations were detailed in a manner that is understandable by experts in the liver physiology and pathology domain, even if they are not M&S experts. An example limitation was that the model was parameterized the same for simulating human-like and rodent-like extents of innervation and only differed by the extent of innervation, which does not use M&S terminology but states the issue in biomedical terms. Note that in the study Verma et al. [26] did not explicitly state the quantitative levels of M&S prediction error arising from the explicitly stated limitations. Under the rubric, those details are not required. There just needs to be sufficient information for an individual to understand under which conditions a model should not be used.

**Rule #5 (Use version control):** The model's conformance was considered to be Extensive (score = 3) as the evolution of the model and the various versions are explicitly documented on GitHub. GitHub is a platform familiar to M&S practitioners but not necessarily to experts in the liver physiology and pathology domain. Hence, the conformance level was not considered Comprehensive (score = 4).

**Rule #6 (Document appropriately):** The model's conformance level is Extensive (score = 3) as comments were provided in the model code, the scope and intended use were described in the main text, and a user guide for M&S practitioners was described in the main text and supplemental files. During the revision process, the model alternative was explained in the main text and an additional supplemental figure was included to detail the results of the model alternative. The user guide was utilized by the independent reviewer (see Rule 8 below) with M&S expertise but little domain familiarity, demonstrating the Extensive level of conformance to Rule 6.

**Rule #7 (Disseminate broadly):** The conformance level was considered as Extensive (score = 3) as the simulations, results and implications were described in the main text and the model files are present in the supplementary material and on GitHub. The software used for this M&S study (Matlab, XPP and a Matlab/XPP interface) are all publicly available either freely or for a fee. The links to these resources and code files were included in the manuscript, enabling potentially Extensive dissemination.

**Rule #8 (Get independent reviews):** The self-assessed model conformance was Extensive (score = 3), as a member of the research group not involved in the study or field performed a review. We note that in order to minimize the bias in the assessment, an internal review, even by a member of the group not involved in the study, is more appropriately scored as a 2 (Adequate). An outside review (outside the primary research groups that conducted the study or even outside the study authors' institutions) could be considered as a 3 (Extensive), and a multi-person independent cross-institutional review, particularly by non-M&S practitioners, could be scored at 4 (Comprehensive).

**Rule #9 (Test competing implementations):** The conformance level only reached a conformance level of Adequate (score = 2) as competing implementations were tested and compared by the first three authors of the paper during the initial manuscript preparation. Furthermore, the competing implementations could only be understood by M&S practitioners familiar with the application domain and the intended context of use, thus justifying the Adequate conformance level. During the manuscript revision stage, the model was further revised, leading to its refinement and improvement. The main text was also updated to include the changes made to the model during revision.

**Rule #10 (Conform to standards):** The model's conformance was considered Adequate (score = 2) as the model was implemented and simulated in a widely used platform for multiscale modeling (Matlab, in this case), along with another freely available and popular software, XPP. Following best coding practices described in Wilson et al. [27] and Hunter-Zinck et al. [28], the model code is commented at critical locations to aid the reader as well. Although the model was documented and disseminated using publicly available online platforms such as GitHub and open access manuscript supplementary material in conformance with rule #7, there was limited evidence of following the operating procedures, guidelines and standards as described in the credible practice of M&S in healthcare: ten rules from a multidisciplinary perspective [9].

The complete self-assessment for this model is included as a supplement to this manuscript (S5 File).

The computational modeling and simulation study of hepatic metabolism has an overall conformance level of Extensive (total numeric score = 30). Therefore, the overall practice of M&S for this biological scenario can be understood by M&S practitioners not familiar with the application domain and intended context of use. For this example, M&S practice to reach a Comprehensive level of conformance to the TR, the authors would need to incorporate additional features into the study. For example, a detailed step-by-step user's and developer's guide such that a non-M&S practitioner can replicate the M&S results would improve the score corresponding to Rule #6. Additionally, the authors could follow a stricter set of operating procedures and guidelines such that the M&S study appropriately conforms to modeling standards in representation, software code and documentation (Rule #10). Lastly, the authors could more comprehensively test and formally document competing implementations of their model for improving the score on Rule #9.

## Discussion

We have described a rubric that specifies detailed criteria for assessing the level of conformance to the Ten Rules for Credible Practice of M&S in Healthcare. The rubric is based on the outreach capability of an M&S practice across a

wide range of stakeholder communities including program leadership, healthcare providers, policy makers and clinical practitioners. The rubric establishes a generalized and graded approach to assess levels of conformance from lowest (Insufficient) to highest (Comprehensive). We have illustrated the application of this rubric in two contexts of use including COVID-19 studies and a liver metabolism model. In the context of assessing COVID-19 studies, we evaluated the consistency of applying the rubric across multiple reviewers. We proposed a scoring scheme that provides a consistent process for M&S assessments and identification of critical credibility conformance gaps across a range of reviewers' familiarity levels. The Ten Rules augmented with the rubric aims to provide a generalized approach for the development and evaluation of the credible practice of M&S in translational and fundamental research endeavors aimed at *in silico* support of systems medicine efforts.

Assessing the outreach capability of an M&S study is useful for those within and outside of a specific scientific discipline. It enables clear communication and application across various stakeholder groups. For example, through the use of the Ten Rules and TR rubric, those working in an industrial setting can easily understand and implement the M&S practices undertaken by the academic research community. Additionally, these parties can communicate to policy makers and higher-level stakeholders that can take action and employ a new development of the Ten Rules and TR rubric to suit their needs. The continuous evolution of the rubric as seen with the implementation of a numerical scoring system for conformance illustrates a framework that is driven by refinement and improvement by the healthcare community.

The TR rubric was introduced to expand the reviewer's utilization of the Ten Rules. Specifically, the introduction of the rubric concept is aimed at expanding the focus of the reviewer from solely evaluating a model based on its validity and accuracy, to including the assessments of how supporting information regarding the M&S credibility engages the community beyond those who are familiar with M&S and the context of use. It is important to note, however, that not every M&S needs to meet a score of Comprehensive to be acceptable. For example, for a Comprehensive conformance level, the outreach is to non-M&S practitioners familiar with the application, while a conformance level of Extensive can be understood by M&S practitioners not familiar with the domain and context of use. Therefore, depending on the use of the model, an Extensive conformance level may be more appropriate than a Comprehensive conformance level.

Assessment of the five discussed COVID-19 model conformances to the Ten Rules shows the value of utilizing such a rubric that prioritizes outreach capability. Specifically, it shows the Ten Rules and TR rubric can establish a cumulative assessment of the TR that has improved consistency in evaluation at each competency level, which was a critical need for decision making support as illustrated by the application to COVID-19 models. Multiple reviewers assessed the conformance of the UPenn COVID-19 model to the Ten Rules. There were notable differences in how the reviewers viewed the supporting credibility evidence, which illustrated that the reviewer's experience level, and their understanding of the context of use relative to the models' intended use, can influence the evaluation. This influence appeared to be nearly orthogonal to the underlying credibility factor domains, leading to the investigation into a more granular and defined TR rubric.

Following the updated rubric application, the consistency of findings between reviewers of similar backgrounds was improved, especially if we consider the consolidated or summed conformance scoring where both reviewers' scores correspond to an overall model conformance level of Adequate.

The assumption that each rule's contribution is equally weighted with respect to the global conformance introduces a limitation in the assessment scheme. For instance, it is possible to accumulate an overall score in the Adequate or Extensive range and still have conformance to one or more individual TR be characterized as Insufficient. This suggests a comprehensive reporting that is more representative of the individual scores may be necessary to communicate the complete M&S credibility outreach picture. An option is to use tailored decision ranking tools such as pairwise comparison and analytical hierarchical processes (AHP) to capture specific community best practice principles by effectively weighting the individual credibility rules. Although the pairwise and AHP approaches may provide domain specific consistency, it is a recommended best practice to provide the set of conformance scores for individual rules as well as the global conformance score when delivering these assessments to decision makers in order to ensure appropriate communications

levels. In this case, the rubric assessed 10 rules that can be grouped into representations tailored for the technical or decision-making community. A proposed method is illustrated in Table 8 representing a grouping of the Ten Rules to derive categorical scores for Development, Application and Supporting Evidence aspects for use in regulatory applications.

The proposed numerical score has its limitations, i.e., a moderate score does not directly inform the intended audience of the individual conformance levels for each of the TR. However, at the high and low extremities, one can readily determine exceptional vs. poor M&S conformance to credible modeling practices using the proposed summative scoring method. A more granular numerical scoring approach can help develop a deeper understanding of the M&S study's conformance level when the summative total numerical score across the TR is in the moderate range. Our alternative approach for a categorized rubric with combined numerical scores for M&S Development (including Rules 1, 3, 5, and 10), M&S Application (Rules 2 and 4), and M&S Supporting Evidence (Rules 6, 7, 8, and 9) strikes a balance. Another alternative is to develop visual representations, for example using radar/spider plots, that can illustrate the multiple levels of conformance across the TR without combining into a single numerical score.

There is additional need and opportunity for streamlining the assessment of M&S activities using the Ten Rules and TR rubric in addition to other associated frameworks. For instance, it may take a significant amount of time to perform the assessment manually. Therefore, automating components of the assessment may provide a capability of assessing the M&S results and associated literature in an unbiased manner. This would be a boon to many communities of practice, especially the healthcare community. A more systematic approach could be taken such that the wider scientific community and stakeholders of the Ten Rules and TR rubric can be included. The Interagency Modeling and Analysis Group (IMAG) and the Multiscale Modeling (MSM) consortium are examples of two groups with significant roles in formulating and developing the Ten Rules and TR rubric. As they both serve a joint purpose of serving the scientific community and providing a collaborative platform to outline good practice of simulation-based medicine, it may be possible to look to their leadership and guidance in systematizing and automating unbiased assessment processes [10].

While a community effort is valuable to progress and implement the ideologies of the Ten Rules and TR rubric, a specific set of guidelines must be established to ensure proper employment. An excellent example of successful first steps in this direction lies with The Physiome scientific journal. The Physiome is an open access journal that, for a small fee, confirms the reproducibility and reusability of the models according to the Ten Rules. By adopting the Ten Rules and TR rubric for M&S credibility, journal curators established that published models generally only conform to an Adequate level of outreach. Implementing an additional guideline in which the model must meet an overall conformance level of Extensive for publication into the journal may promote the benefits of M&S outreach capability to the scientific community.

Future implementations of the Ten Rules and TR rubric could consider how the credible practice for assessment of an M&S practice may be transferable from one context to another. For example, another context of use was noted for the model of liver metabolism. However, the reviewers did not assess the model in this alternate context. It is an open question as to how the assessments of the Ten Rules can be applied to the alternate contexts of use and under what conditions this can occur. It may be the case where the previous assessments of only some of the Rules can be transferred

**Table 8. Example of categorizing the TR and rubric assessments to support regulatory applications according to Reviewer 1's assessment of the UPenn COVID-19 model's conformance to the Ten Rules.**

| Development | | Application | | Supporting Evidence | | Overall |
|---|---|---|---|---|---|---|
| Rule 1 | 4 | Rule 2 | 2 | Rule 6 | 3 | Summary Score 2.1 |
| Rule 3 | 0 | Rule 4 | 2 | Rule 7 | 4 | |
| Rule 5 | 3 | | | Rule 8 | 0 | |
| Rule 10 | 2 | | | Rule 9 | 1 | |
| Development Score 2.25 | | Application Score 2 | | Supporting Evidence Score 2 | | |

while others may be "non-transferable". Additionally, M&S practices may be altered at different user levels. For instance, it may not be appropriate to use a model built on data from a local hospital system and apply the model at the national scale. Furthermore, the transferability issue has implications for assessing the conformance of ensemble models, or a single model that contains multiple diverse models, to the Ten Rules. One potential solution may be to provide reasoning for including each of the models into the greater ensemble model in the same way that a single equation within an ordinary differential equation (ODE)-based model would be explained.

In an additional future implementation, we propose that the current rules and rubric can be adjusted to more explicitly account for patient-specific/digital twin models as they begin to be utilized in the clinical setting. An updated and extended set of rules and practices may be developed for assessing and ensuring the credibility of these models. The need for an updated list of rules is essential in the personalized modeling realm as the current methods lack consistency and credibility, especially within the clinic. Additionally, the Ten Rules may not be adequate in assessing the complexity and detail required for digital twin modeling. The updated and extended rules for digital twin modeling can then be used as a guide during the developmental stages of model development to avoid the problems seen with current digital twin efforts as previously discussed. Future modeling efforts that are guided by future implementations of the rules may also establish more trust and interaction between the modeler and clinician, thereby bridging the gap that currently exists in translating computational models from research into the healthcare field.

The TR rubric is used to assess a model's conformance to the Ten Rules for credible practice in M&S in healthcare. It is highly recommended that the M&S activity in the healthcare domain reaches either a conformance level of Comprehensive or Extensive. Both conformance levels have their own intended outreach capability as Comprehensive models can be understood by **non**-M&S practitioners familiar with the application domain and the intended context of use while Extensive models can be understood by M&S practitioners **not** familiar with the application domain and the intended context of use. Thus, defining which group must use the M&S results to support their decision is of utmost importance. The outreach goal for a given model is to be as clear and comprehensible to as broad an audience as possible such that the model can be widely adopted.

In conclusion, we formulated a rubric that promotes consistent and continuous evolution and testing of M&S practices such that one can reach the appropriate outreach level. In addition to the evolution of individual models, the TR rubric may evolve to meet the needs of its users as one continues to test its conformance to the Ten Rules. The development of the TR rubric has initiated a large community effort to assess the outreach, reproducibility, replicability, and credibility of M&S studies in the scientific healthcare domain.

## Supporting information

**S1 File. The extended Ten Simple Rules rubric with detailed criteria.**
(XLSX)

**S2 File. Reviewer 1's complete assessment and conformance testing to the Ten Simple Rules using the rubric applied to the COVID-19 modeling studies.**
(DOCX)

**S3 File. Reviewer 2's complete assessment and conformance testing to the Ten Simple Rules using the rubric applied to the COVID-19 modeling studies.**
(DOCX)

**S4 File. Reviewer 3's complete assessment and conformance testing to the Ten Simple Rules using the rubric applied to the COVID-19 modeling studies.**
(XLSX)

**S5 File. The complete self-assessment and conformance testing to the Ten Simple Rules using the rubric applied to the Verma et al. (26) multi-scale, multi-organ model of hepatic metabolism.**
(DOCX)

## Acknowledgments

The authors would like to acknowledge the Interagency Modeling and Analysis Group and the Multiscale Modeling Consortium, who enabled activities of the Committee on Credible Practice of Modeling & Simulation in Healthcare. We would also like to thank the individual contributions of the Committee members who continue to help advance the efforts of the Committee but were not able to contribute to this manuscript.

## Author contributions

**Conceptualization:** Rajanikanth Vadigepalli, Ahmet Erdemir, Lealem Mulugeta, Jerry G. Myers.

**Funding acquisition:** Rajanikanth Vadigepalli, Alexandra Manchel, Ahmet Erdemir, Joy P. Ku, William W Lytton.

**Investigation:** Rajanikanth Vadigepalli, Alexandra Manchel, Ahmet Erdemir, Lealem Mulugeta, Bruno V. Rego.

**Methodology:** Rajanikanth Vadigepalli.

**Writing – original draft:** Rajanikanth Vadigepalli, Alexandra Manchel.

**Writing – review & editing:** Rajanikanth Vadigepalli, Ahmet Erdemir, Lealem Mulugeta, Joy P. Ku, Bruno V. Rego, Marc Horner, William W Lytton, Jerry G. Myers.

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
