## [Decision Letter · Decision Letter 0]

Dear Dr. Vadigepalli,

Thank you for submitting your manuscript to PLOS ONE and your patience as we had significant difficulties identifying and contacting appropriate reviewers. After careful consideration, we feel that it has merit but does not fully meet PLOS ONE’s publication criteria as it currently stands. Therefore, we invite you to submit a revised version of the manuscript that addresses the points raised during the review process.

The reviewers were interested in the manuscript and its potential utility for readers, but each has identified critical aspects of the narrative that the authors must address. Clarity, with concision, are the main ideas behind both reviewers' comments. Reviewer 2 in particular has identified specific attributes of the narrative that require careful attention by the authors to achieve the requisite level of clarity of the intended purpose of the rubric, and specifically for supporting coherent decision making that is based on (wholly or in part) the rubric. Please address all reviewer comments, whether they specify a revision or suggestion is "minor" or "major". I agree with both reviewers that clarity and coherence, for any reader and ultimately user of the rubric, are essential. 

Thank you again for your patience and for submitting this manuscript. We look forward to receiving your revised manuscript soon.

Kind regards,

Rochelle E. Tractenberg, PhD, MPH, PhD, FASA, FAAAS, FISI

Section Editor

PLOS ONE

Journal Requirements:

[I have read the journal's policy and the authors of this manuscript have the following competing interests: Ahmet Erdemir owns and operates innodof, LLC, a consulting company for modeling and simulation. Lealem Mulugeta owns and operates InSilico Labs LLC and Medalist Performance. InSilico Labs provides computational modeling and simulation products and services, and Medalist Performance applies computational and biomedical approaches to provide peak performance coaching services to tactical professionals, athletes, astronauts, and executives. Marc Horner is employed by ANSYS, Inc., a company that develops commercial off-the-shelf computational modeling software. The remaining authors have declared that no competing interests exist.].

Please confirm that this does not alter your adherence to all PLOS ONE policies on sharing data and materials, by including the following statement: ""This does not alter our adherence to PLOS ONE policies on sharing data and materials.” (as detailed online in our guide for authors http://journals.plos.org/plosone/s/competing-interests ). If there are restrictions on sharing of data and/or materials, please state these. Please note that we cannot proceed with consideration of your article until this information has been declared.

Please confirm at this time whether or not your submission contains all raw data required to replicate the results of your study. Authors must share the “minimal data set” for their submission. PLOS defines the minimal data set to consist of the data required to replicate all study findings reported in the article, as well as related metadata and methods (https://journals.plos.org/plosone/s/data-availability#loc-minimal-data-set-definition ).

If your submission does not contain these data, please either upload them as Supporting Information files or deposit them to a stable, public repository and provide us with the relevant URLs, DOIs, or accession numbers. For a list of recommended repositories, please see https://journals.plos.org/plosone/s/recommended-repositories .

Reviewers' comments:

Reviewer's Responses to Questions

**Comments to the Author**

1. Is the manuscript technically sound, and do the data support the conclusions?

Reviewer #1: No

Reviewer #2: Partly

2. Has the statistical analysis been performed appropriately and rigorously?

Reviewer #1: N/A

Reviewer #2: N/A

3. Have the authors made all data underlying the findings in their manuscript fully available?

Reviewer #1: Yes

Reviewer #2: Yes

4. Is the manuscript presented in an intelligible fashion and written in standard English?

Reviewer #1: Yes

Reviewer #2: Yes

Reviewer #1: Overall, I really think you are onto something here. I can see hints of how something like what you are talking about can be very useful and how it may actually side step some of the issues that others have fallen into when trying to provide similar information about models and simulations. For example, what you are talking about here has been tried by the national labs with PCMM. Only it didn't work. I think they even revised it 4 or 6 times, and they got it to a good point, but it never could provide that "single number" that everyone wants.

However, I ultimately got very confused about the purpose of the rubric. Is it a measure of the the credibility of a model/simulation or is it a measure of the availability and "understandability" of the information needed to determine the credibility of a model/simulation?

Consider two different simulations. For the first simulation, assume I have followed Ten Rules to such an extent than anyone who saw my work would immediately agree that I clearly followed all Ten Rules. However, assume that for this simulation I am not providing ANY information such that someone else could independently determine that I followed any of the rules. Would the rubric rate this scenario high or low?

Conversely, consider a situation in which I followed none of the rules. However, I have provided the information such that everyone can quickly determine that I have followed none of the Ten Rules. Would the rubric rate this scenario high or low?

I feel like "large changes" are needed to this paper, not because a lot of wording needs to change. Overall, the paper is well written. However, I feel like what the paper is saying needs to be made much clearer. I think this can be done with a few wording changes, but I think it is extremely important as those wording changes define the rubric, and right now I do not understand the rubric at all.

Just to re-iterate, even though I say "major revision", I don't expect to see that many wording changes in the paper, but there needs to be a further definition of just what the rubric is, and since that is the heart of the paper, that is a "major change" to me.

Reviewer #2: The manuscript “A rubric for assessing conformance to the Ten Rules for credible practice of modeling and simulation in healthcare” by Vadigepalli et al describes a process for assessing the outreach capability of modeling and simulation documentation based on each of the previously established Ten Rules. In brief, outreach is scored from level 0 (“Insufficient”) to level 4 (“Comprehensive”). An aggregate score is also calculated by summing the scores, resulting in an overall score between 0 and 40. The rubric is demonstrated using COVID-19 modeling case studies from literature. Although assessment of individual Rules based on outreach capability differed among recruited review participants, aggregate scores mostly agreed within a few points, supporting the robustness of the rubric for assessing outreach (e.g., Table 7).

That said, I have significant concerns regarding language used to describe the rubric and potential for miscommunication and misinterpretation when used in practice.

Major Comments:

• Ten Rules Conformance versus Ten Rules Outreach Potential:

The purpose of the rubric requires explicit clarification. Two concepts are presented throughout the manuscript: 1) *technical* conformance to the Ten Rules, and 2) *outreach* potential of modeling study documentation based on each of the Ten Rules. From this reviewer’s perspective, the two concepts are orthogonal and should not be presented together. For example, a model with excellent predictive accuracy that conforms to the Ten Rules may not have supporting plain-English documentation, limiting outreach. Conversely, a terrible model with limited to no predictive accuracy could have very clear documentation that facilitates outreach. Other permutations are likewise possible (high conformance and outreach, low conformance and outreach).

However, several times throughout the manuscript, the concepts of Ten Rules conformance and outreach potential are intermingled. Line 109 of the Introduction explains the purpose of the proposed rubric:

“To address this issue, the Committee formulated a rubric based on the *ability of the M&S to facilitate outreach of the results to a wide range of stakeholders* from context-specific M&S practitioners to policymakers.” (emphasis added)

Elsewhere, the text implies the rubric assesses *conformance* to the Ten Rules. The title itself states: “A rubric for assessing *conformance* to the Ten Rules for credible practice…”.

The disconnect is perhaps more clearly demonstrated on lines 441-445:

“Rule #5 (Use version control): The model’s conformance was considered to be Extensive (score = 3) as the evolution of the model and the various versions are explicitly documented on GitHub. GitHub is a platform familiar to M&S practitioners but not necessarily to experts in the liver physiology and pathology domain. Hence, the conformance level was not considered Comprehensive (score = 4).”

Here, the score is lowered *not* because of lack of conformance to the Ten Rules’ Rule #5 for version control, but because of the perceived difficulty in communication and outreach without an accompanying explanation of git. Had the documentation included lay explanations of version control, but version control was poorly executed, could the highest outreach score of “Comprehensive” (5) have been achieved? This is potentially problematic. Additional excerpts that may be similarly confusing to readers are provided below.

As presented, the outreach rubric could lead decision makers interpreting scores to believe an inaccurate model is highly credible solely because the outreach capability score is high. This is potentially dangerous if decision makers are not provided clear context of the rubric and its purpose.

To resolve the concern, the reviewer recommends revising the manuscript to consistently present the scope of the rubric as “A rubric for assessing *outreach capability*” or similar and omitting or isolating mentions of technical conformance to the Ten Rules and model credibility. The scope of the outreach rubric should also be emphasized throughout the manuscript to avoid confusion for readers, including in the manuscript title.

• Introduction Length: The Introduction as drafted is long (eight paragraphs, 5 pages). The manuscript would benefit from a more concise Introduction to summarize relevant background and motivate the need for the outreach rubric specifically.

• Additional excerpts related to rubric scope concern:

Line 34: “Here, we present a pragmatic rubric for assessing the conformance of an M&S activity to the TR”: Per the above, this does not appear to be true. Revise to clarify the scope of the rubric is limited to *outreach capability* (or similar wording).

Line 38: “We used the rubric to evaluate the *conformance* of two computational modeling activities” -> “We used the rubric to evaluate the *outreach capability* of two computational modeling activities”

Line 46: “Our findings demonstrate that the TR rubric represents a systematic tool for assessing the *conformance* of an M&S activity” -> “…assessing the *outreach capability* of…”

Line 233: “The rubric assesses the conformance to each of the Ten Rules based on the extent of outreach to each group”: Can a model conform to the Ten Rules but be poorly presented for outreach? Recommend rephrasing to omit mention of conformance.

See also Lines 137, 145, 152, Table 2, Figure 1 caption, 174, etc (135 total uses of the word “conformance”).

**Do you want your identity to be public for this peer review?** For information about this choice, including consent withdrawal, please see our Privacy Policy

Reviewer #1: **Yes: ** Joshua S Kaizer

Reviewer #2: No

---

## [Author Response · Author response to Decision Letter 1]

24 Mar 2025

Please see the uploaded Response to Review document.

---

## [Decision Letter · Decision Letter 1]

A rubric for assessing conformance to the Ten Rules for credible practice of modeling and simulation in healthcare

PONE-D-24-43327R1

Dear Dr. Vadigepalli,

We’re pleased to inform you that your manuscript has been judged scientifically suitable for publication and will be formally accepted for publication once it meets all outstanding technical requirements. Thank you very much for your patience with the process.

Kind regards,

Rochelle E. Tractenberg, PhD, PhD, MPH, FASA, FAAAS, FISI

Section Editor

PLOS ONE

Additional Editor Comments (optional):

Reviewers' comments:

Reviewer's Responses to Questions

**Comments to the Author**

Reviewer #1: All comments have been addressed

2. Is the manuscript technically sound, and do the data support the conclusions?

Reviewer #1: Yes

3. Has the statistical analysis been performed appropriately and rigorously?

Reviewer #1: N/A

4. Have the authors made all data underlying the findings in their manuscript fully available?

Reviewer #1: Yes

5. Is the manuscript presented in an intelligible fashion and written in standard English?

Reviewer #1: Yes

Reviewer #1: I still have certain questions, but I have those same questions for other published frameworks. Overall, I think you have made a valuable contribution, and congratulate you on the effort.

**Do you want your identity to be public for this peer review?** For information about this choice, including consent withdrawal, please see our Privacy Policy

Reviewer #1: **Yes: ** Joshua S. Kaizer

---

## [Editor Report · Acceptance letter]

PONE-D-24-43327R1

PLOS ONE

Dear Dr. Vadigepalli,

I'm pleased to inform you that your manuscript has been deemed suitable for publication in PLOS ONE. Congratulations! Your manuscript is now being handed over to our production team.

Kind regards,

on behalf of

Professor Rochelle E. Tractenberg

Section Editor

PLOS ONE